# Insights into Grape Ripe Rot: A Focus on the *Colletotrichum gloeosporioides* Species Complex and Its Management Strategies

**DOI:** 10.3390/plants12152873

**Published:** 2023-08-04

**Authors:** Ting-Fang Hsieh, Yuan-Min Shen, Jin-Hsing Huang, Jyh-Nong Tsai, Ming-Te Lu, Chu-Ping Lin

**Affiliations:** 1Plant Pathology Division, Taiwan Agricultural Research Institute, Taichung City 41362, Taiwan; tfhsieh@tari.gov.tw (T.-F.H.); jhhuang@tari.gov.tw (J.-H.H.); tsaijn@tari.gov.tw (J.-N.T.); 2Master Program for Plant Medicine, College of Bio-Resources and Agriculture, National Taiwan University, Taipei 10617, Taiwan; shenym@ntu.edu.tw; 3Crop Science Division, Taiwan Agricultural Research Institute, Taichung City 41326, Taiwan; mingtelu@tari.gov.tw

**Keywords:** grape ripe rot, *Colletotrichum gloeosporioides* species complex, infection process, *C. viniferum*, disease management

## Abstract

Grape ripe rot, which is predominantly caused by the *Colletotrichum* species, presents a growing threat to global grape cultivation. This threat is amplified by the increasing populations of the *Colletotrichum* species in response to warmer climates. In this review, we investigate the wide-ranging spectrum of grape ripe rot, specifically highlighting the role and characteristics of the *C. gloeosporioides* species complex (CGSC). We incorporate this understanding as we explore the diverse symptoms that lead to infected grapevines, their intricate life cycle and epidemiology, and the escalating prevalence of *C. viniferum* in Asia and globally. Furthermore, we delve into numerous disease management strategies, both conventional and emerging, such as prevention and mitigation measures. These strategies include the examination of host resistances, beneficial cultivation practices, sanitation measures, microbiome health maintenance, fungicide choice and resistance, as well as integrated management approaches. This review seeks to enhance our understanding of this globally significant disease, aspiring to assist in the development and improvement of effective prevention and control strategies.

## 1. Introduction

Grapes (*Vitis* spp.) are extensively cultivated worldwide, and they have considerable global importance and economic impact. The global vineyard area was estimated to be approximately 7.3 Mha in 2021. Over half of worldwide grape production contributes to the winemaking industry, with the remainder mainly used as table grapes, dried grapes, and the production of musts and juices [1]. However, this substantial industry encounters significant threats from fruit diseases that affect the grape’s berries, such as bitter rot, black rot, Botrytis bunch rot, and—notably—ripe rot [2,3].

Ripe rot is particularly adapted to warm, humid, subtropical climates, and it poses a significant threat to grape cultivation, especially across South and North America, Australasia, and Asia—including Brazil, the United States, Australia, Taiwan, Japan, Korea, and China [4,5,6,7,8,9,10]. It has been responsible for losses exceeding 30%, and, in some cases, up to 60% or even more [4,11,12]. This disease not only reduces grape yields, but also adversely affects the chemical composition and quality of grapes and wine, leading to off flavors and a brownish color [13,14,15,16].

In this review, we provide an overview of grape ripe rot that is caused by the complex of *Colletotrichum* spp. fungi, with a particular emphasis on the *Colletotrichum gloeosporioides* species complex (CGSC). We examine its symptoms, infection factors, and the emerging pathogen within the complex. Additionally, preventive and suppressive strategies that can be utilized for the integrated management of grape ripe rot are extensively discussed.

## 2. Grape Ripe Rot Caused by the *Colletotrichum* Complex

*Colletotrichum*, recognized for its role in causing ripe rot in grapes and for affecting many other plants, is ranked among the top ten plant fungal pathogens [2,17,18]. *C. gloeosporioides* (Penz.) Penz. & Sacc. and *C. acutatum* J.H. Simmonds ex J. H. Simmonds are the major species within this context [18,19]. The early classification of the *Colletotrichum* species primarily relied on features such as colony morphology, conidial shape and size, appressoria, physiological characteristics, and the host plant [20,21]. This led to significant ambiguity, as some strains identified as the same species based on the morphology exhibited, or due to different pathogenicity or physiological characteristics; thus, this made *Colletotrichum* a catalog of confusion [17,22].

Since 2012, the introduction of multilocus sequence analysis (MLSA) has marked a prominent development in the field. MLSA employs an array of loci, including *act*, *chs-1*, *gadph*, *tub2*, *his3*, *cal*, *tef*, *gs*, *sod2*, and ITS, among others, for delineating species within this genus. This approach has facilitated the reclassification of the genus into at least 15 complexes, encompassing a total of 257 species [17,23,24,25]. These include the CGSC, the *C. acutatum* species complex (CASC), and others. Moreover, the *ApMat* locus demonstrated notable utility in distinguishing species within the CGSC, even when used alone [26,27]. Notably, pre-2012 studies (which often lack multigene analysis) should be interpreted with caution. When referring to a species complex such as *C. gloeosporioides* and *C. acutatum* without clear molecular evidence, the term *sensu lato* (*s.l.*; in a broad sense) is usually included for clarification. On the other hand, *sensu stricto* (*s.s.*; in a narrow sense) is used for the species that have been identified through MLSA or *ApMat* marker analysis.

First identified in the United States in 1891, grape ripe rot was originally linked to *C. gloeosporioides s.l.* [28]. As research progressed, *C. acutatum*
*s.l.* was also found to be a potential causative agent of this disease [6,7,10,29]. Today, grape ripe rot is understood to be triggered by a blend of the *Colletotrichum* species, predominantly from the CGSC and CASC, with occasional involvement from the *C. boninense* and *C. orchidearum* species complexes (Table 1).

The CGSC and CASC demonstrate significant divergence in multiple aspects, including not only temperature requirements, infection rates, spore dispersal, and fungicide sensitivity [6,45,46,47,48], but also geographical prevalence. For instance, various CASC members that are implicated in grape rot in America and Australia are notably absent in Asia, while *C. viniferum* L.J. Peng, L. Cai, K.D. Hyde & Z.Y. Liu, (a member of the CGSC), causes severe infections in parts of South America and Asia, but it is scarcely documented in other regions (Table 1). From a geographical perspective, CGSC is commonly found in warmer climates and CASC in cooler environments [12,19,49].

In the context of climate change and rising temperatures, regions previously unaffected by grape ripe rot may increasingly be confronted by the disease. Specifically, the warm-climate-preferring CGSC might become more prevalent in regions experiencing elevated temperatures. Furthermore, areas previously free of this disease could start facing encounters with ripe rot, either influenced by CGSC or induced by the cooler-environment-preferring CASC. Despite acknowledging the crucial role of both CGSC and CASC in the complex nature of grape ripe rot, this review primarily focus on CGSC owing to its significant influence in subtropical regions and its broader global distribution.

## 3. Diverse Grapevine Symptoms Caused by the CGSC

A wide array of symptoms on berries, flowers, and other vegetative tissues of grapevine can be attributed to species within the *Colletotrichum* genus and CGSC depending on their virulence and complicated interactions with various factors. 

Ripe rot typically manifests as a berry-rot type, beginning with lesions that evolve into dark, sunken necrotic spots with concentric rings that produce acervuli (asexual fruiting bodies of the pathogen). Upon exposure to moist conditions, the lesion surfaces proliferate into orange- or salmon-colored conidial masses, and ultimately result in the drying and mummification of decaying berries (Figure 1).

While the symptoms on berries are relatively similar across different species that cause ripe rot [7,9,38,50], there is significant variation in the symptoms that manifest on other parts of the grapevine, such as flowers, leaves, and canes. Certain species predominantly affect berries and usually do not cause noticeable symptoms on most infected grape parts. These species include *C. fructicola* Prihast., L. Cai & K. D. Hyde [35,51], *C. gloeosporioides s.s.* [51], and *C. tropicale* E. I. Rojas, S. A. Rehner & Samuels [35], as well as *C. fioriniae* Marcelino & Gouli and *C. nymphaeae* (Pass.) Aa (which is a member of the CASC [12,51]). Conversely, some members of the CGSC can also cause a multipart rot of the grapevine apart from the berries alone, such as black spots, blight, and canker on other grape parts like flowers, leaves, and other vegetative tissues (Figure 2). Notable examples include *C. aenigma* [32], *C. viniferum* [8,32,39], and other *C. gloeosporioides s.l.* [7,52,53], which have yet to be conclusively identified through rigorous molecular analysis. Interestingly, although this has not been consistently observed, there are occasional reports of *C. gloeosporioides s.s.* causing localized leaf lesions [54], and *C. viniferum* causing necrotic lesions similar to the hypersensitive reaction on leaves [51]. *C. siamense* has been established as one of the causative pathogens of ripe rot in the table grapes variety *V. vinifera* ‘BRS Vitória’ in Brazil [36]. However, according to our unpublished data, *C. siamense* (which is occasionally isolated from necrotic young grape berries), showed no pathogenicity on the berries of the hybrid variety *V. labrusca* × *vinifera* cv. Kyoho. There is a range of literature on *C. siamense*; some reports indicate leaf infections [32], while others describe it as a saprophyte on grapes [51,55]. These variations might be attributed to the intricate interplay between grape cultivars, the differences in virulence of the causal strain or CGSC, and even environmental factors [5,9,30,34]. Other species within the CGSC, such as *C. conoides* and *C. temperatum*, have also been suggested to be linked to grape ripe rot, though their pathogenicity on berries remains uncertain [4,12].

An intriguing exception within the naming convention of grape diseases related to *Colletotrichum* spp. is worth underscoring. In most plants, “anthracnose” often refers to diseases caused by *Colletotrichum* spp. [56,57,58]. However, in grapevine, “anthracnose” is specifically designated for a disease caused by *Elsinoe ampelina* (de Bary) Shear, which exhibits symptoms distinct from the classic ripe rot. These manifest as sunken necrotic lesions with grayish centers and brownish margins on berries, stems, and shoots, as well as small dead areas on leaves that lead to irregular holes [2,59]. Interestingly, certain *Colletotrichum* species have been reported to induce symptoms that are similar to the anthracnose caused by *E. ampelina*. Examples include *C. gloeosporioides s.l.* in India and the Philippines [60,61], along with species of other complexes such as *C. acutatum s.l.* and *C. capsici s.l.* in India [62,63], and *C. fioriniae* (CASC) in the U.S. [64].

The diversity of symptoms caused by *Colletotrichum* on grapevine highlights the critical necessity of precise identification in pursuing efficacious disease management, especially considering the different effects on the grape tissues across the *Colletotrichum* species. The berry-rot type of the grape rot disease caused by CGSC will particularly be further discussed in greater detail in the subsequent sections.

## 4. CGSC Life Cycle and Infection Factors

Gaining insights into the complex interactions between environmental factors, pathogens, and hosts is crucial for predicting the disease occurrences in grapevines. As the epidemiology and life cycle of the CGSC are explored, a deeper understanding of these intricate relationships will contribute to our overall knowledge of plant–pathogen interactions and their influence on grapevine health.

### 4.1. Pathogen

#### 4.1.1. Lifestyles and Infection Processes

*Colletotrichum* species share similar, although not identical, lifestyles that are influenced by pathogen species, the disease resistance of the host tissue and its physiological maturity, as well as environmental conditions [65]. Nevertheless, the diversity of the symptoms observed on the grapevine organs other than on the berries also reflect the variety in their lifestyles within different parts of the grapevine.

During berry infection, histopathological studies have generally shown that some *C. gloeosporioides s.l.* exhibit a hemibiotrophic lifestyle [66,67]. As conidia germinate on grape berry surfaces, appressoria form—which are positively correlated with grape rot disease severity—produce penetration pegs and enzymes, as well as breach the cuticle within a week after inoculation [66,67,68]. Fungal growth halts until veraison and then resumes inter- and intracellularly, ultimately leading to cellular collapse, necrosis, and ripe rot symptoms that develop as mature acervuli emerge [66]. Similarly, when *C. viniferum* was inoculated on pea-sized berries, the symptoms developed only after veraison [8], which also suggests a latent infection and the possibility of a hemibiotrophic lifestyle during berry infection.

In regard to the infections on grapevine organs other than on berries, species such as *C. fructicola*-like and *C. aenigma* isolates do not induce symptoms. However, they do produce acervuli on blooms, suggesting a biotrophic lifestyle or endophytic behavior on asymptomatic flowers [12]. In contrast, *C. viniferum* causes severe necrosis on blooms, and undergoes significant secondary conidiation, indicating a primarily necrotrophic lifestyle [8].

#### 4.1.2. Overwintering Structures and Primary Inoculum Sources

Several studies have demonstrated that CGSC members can overwinter in various grape tissues such as nodes, tendrils, pedicels, or peduncles, as well as debris, such as mummified berries and necrotic or desiccated leaves [8,69,70]. Additionally, sclerotia have been observed in some *C. viniferum* cultures [9], suggesting sclerotium as a possible overwintering structure in the field.

Under favorable conditions for sporulation, water-dispersed conidia have the potential to be carried onto new tissues such as flowers, tendrils, leaves, or young berry clusters. Once there, the conidia germinate and either induce advanced symptoms or establish quiescent or latent infections [8,12,35,52,70,71] depending on the species and the interaction between the hosts. 

Furthermore, perithecia or ascospores, which are potentially spread by wind, have been observed in CGSC species such as *C. viniferum* and others [6,9,53]. Observations of the potential wind-mediated dispersal of CGSC members have also been reported [4]. Collectively, this evidence indicates that the causal CGSC members of ripe rot may employ both water and wind as vectors for dissemination, subject to environmental conditions.

#### 4.1.3. Secondary Inoculum Sources and Infection Dynamics

Compared to primary inoculum, secondary inoculum are more closely correlated with disease severity for polycyclic pathogens such as *Colletotrichum* spp. [72,73]. As different species can cause varying levels of signs on the plant parts, the location and quantity of secondary inoculum may also vary.

*Colletotrichum* spp. can undergo secondary conidiation on asymptomatic tissues; for instance, *C. fructicola*-like isolates can produce conidia on unblemished flowers [12]. Even more notably, *C. viniferum* have been observed producing conidia on necrotic flowers. When blooming inflorescences are infected with *C. viniferum*, the symptoms can involve necrosis on various parts, such as rachises, subrachises, and flowers, with the flower cap (calyptra) being the most susceptible (unpublished data). The infected deciduous calyptras, which carry abundant conidia, are very lightweight and can be blown in any direction, potentially leading to secondary infections.

Although the infection pattern of individual CGSC members during the host bloom stage is yet to be comprehensively understood, observations indicate that flowers infected by CASC species (*C. acutatum s.l.*) result in an increase in disease incidence in the subsequent berry clusters [74,75]. In contrast, Cosseboom and Hu (2022) [4] suggest that infection at the bloom stage may not be as crucial as it is in the fruit stage.

### 4.2. Host: Susceptibility of Berries to CGSC Infection

Various studies have examined the susceptibility of grape berries to CGSC infection at different physiological stages. Daykin (1984) [69] found that berries are equally susceptible to *C. gloeosporioides s.l.* at all stages, while Fukaya (2001) [70] reported susceptibility until the stone-hardening stage. In contrast, Cosseboom and Hu (2022) [4] observed the ontogenic susceptibility of berries in natural infections by *Colletotrichum* spp. that consisted primarily of the CASC and a few CGSC members, with the stage after veraison being significantly susceptible. Nonetheless, the variability in berry susceptibility across the different studies can be attributed to several factors. These may encompass the particular species of *Colletotrichum* involved, the resistance levels of grape cultivars, and the complex interactions between grape cultivars and specific CGSC species [29].

### 4.3. Environmental Factors

The impact of environmental factors on grape berry infection by the artificial inoculation of *C. gloeosporioides s.l.* has shown that optimal infection conditions arise when the temperature is within the 25–30 °C range, and when there is a minimum wetness duration of 8 h [68]. A long-term vineyard monitoring spanning 12 years revealed that conidia from overwintered inoculum were released when three consecutive days exhibited temperatures above 15 °C, mean minimum temperatures that exceeded 10 °C, and a total precipitation surpassing 10 mm during that period [70].

Additionally, rainfall influences leaf wetness duration and strongly correlates with the dispersal of *C. gloeosporioides s.l.* conidia [69,70]. Ji et al. (2021) [71] developed a simulation model that considered the dynamic influence of weather on the epidemiology of the grape ripe rot caused by *Colletotrichum* spp. The model predicted disease occurrence by incorporating the effects of rainfall, temperature, and host susceptibility on fungal infection and sporulation processes. The accuracy of the prediction model was validated using field data, demonstrating its potential for supporting decision making in vineyard management practices and in improving the timing of fungicide applications. Cosseboom et al. (2022) [11] developed a comparable model, but with a focus on *C. fioriniae*, which is a CASC member. Nevertheless, it is important to highlight the distinctions between the CGSC and CASC species in terms of infection strategies, environmental conditions, and tissue preferences, as was discussed above. Therefore, despite the similarities in their general life cycles, studies focusing on the predominant CASC species need to be carefully adapted to understand the CGSC.

## 5. Changes in the Primary Causal Species

It is essential to recognize that dominant species can undergo shifts over time due to factors such as environmental conditions, host resistance, and fungicide application. Additionally, climate change and human activities potentially play influential roles, thus underscoring the importance of sustained research in this domain [76,77,78]. In India, climate change has been implicated in *C. gloeosporioides s.l.* replacing *E. ampelina* as the dominant pathogen causing anthracnose in grapevines [79].

The *Colletotrichum* species that causes ripe rot in grapes constitutes a diverse group of fungal pathogens. Given the complexity and diversity of their infection dynamics as mentioned above, it becomes imperative to understand the population dynamics of these species to devise effective strategies for maintaining the health of grapevines. Initially, in Taiwan, the CGSC species that caused no symptoms on leaves, likely *C. fructicola*, were postulated to be the prevalent pathogens for grape ripe rot [35]. However, a shift is evident in recent studies, where *C. viniferum*, a species within the CGSC that possesses different and stronger virulence toward berries and other grapevine parts, has emerged as the predominant pathogen [8,80].

## 6. *C. viniferum* Emergence in Asia and Other Regions

Grape ripe rot caused by *C. viniferum* is widely reported in Asian regions such as China, Japan, Korea, and Taiwan, as well as in South American countries such as Brazil. A related yet distinct pathogen, “Clade V”, a *C. viniferum*-like species that is identified through phylogenetic analysis through using the *ApMat* marker, has also been found. *C. viniferum* or Clade V is the primary pathogen causing grape ripe rot in most grape-producing regions in these areas (Table 1). 

*C. viniferum* was first recorded in mainland China [9], as evidenced by a phylogenetic tree constructed using MLSA (*act*, *chs-1*, *gapdh*, *tub2*, and ITS). In a separate study, it was suggested that *C. viniferum* may represent a cryptic species complex comprising distinct lineages [5,30]. Moreover, the research conducted by Lin et al. (2022) [8] highlights the significant role of geographical separation in the population differentiation of *C. viniferum* that occurs among Taiwan, China, and Brazil. 

Generally, *C. viniferum* displays a greater virulence to berries than other pathogens within CGSC and CASC [5,30,34,35,38]. As previously highlighted, *C. viniferum* has the ability to infect multiple parts of the grapevine. This not only increases its potential range of infection sites, but also contributes to its enhanced virulence compared to most *Colletotrichum* spp. identified to date. The host range of *C. viniferum* has been expanding, with new hosts identified including strawberry [81], walnut [81,82], chili [83] in China, pomegranate in India [84], and *Hopea odorata* [85] in Bangladesh.

Despite limited research on the molecular mechanisms underlying the infection of grapes by *Colletotrichum*, the recent release of the *C. viniferum* genome sequence marks a significant advancement in the field [86]. The transcriptome analysis of grape berries resistant to *C. viniferum* has also provided valuable insights, revealing that pathogen invasion disrupts calcium levels and activates the MAPK pathway, subsequently upregulating transcription factors such as WRKY, ERF, and MYB [87]. This research highlights the importance of understanding disease resistance responses, which include the accumulation of protective compounds, such as stilbene phytoalexins and anthocyanins, the expression of plant–pathogen interaction genes, and alterations in metabolism, such as in peroxisomes and fatty acids.

## 7. Management of Grape Ripe Rot Caused by the CGSC

Managing the plant diseases caused by *Colletotrichum* poses a complex challenge due to the diverse species complexes and the multidimensional aspects of interactions between the host, pathogen, and environment [19]. In this section, the discussion follows the order of principles of integrated pest management that was outlined in Barzman et al. (2015) [88], and the aim is to review the principles specifically related to the management of grape ripe rot caused by the CGSC. Disease prevention and suppression strategies can be achieved through the use of resistant cultivars, the implementation of rain sheltering and bagging techniques, the enhancement of beneficial organisms, the utilization of non-chemical measures, the selection of suitable fungicides, and reductions in fungicide resistance.

### 7.1. Cultivars with Resistance to Ripe Rot

Different grapevine species and cultivars show varying levels of resistance to grape ripe rot disease [44,89,90,91,92]. Some wild oriental *Vitis* species have been observed to be completely resistant to the CGSC, including certain genetic materials of *V. amurensis* Rupr., *V. heyneana* Roem. & Schult. (*V. quinquangularis* Rehder), and *V. davidi* (Rom.Caill.) Foëx [81]. Other plants may exhibit lower disease incidences than susceptible cultivars, such as *V. bryoniifolia* cv. Bunge (*V. adstricta* Hance), *V. piasezkii* cv. Maxim., and *V. romanetii* cv. Rom.Caill. [81]. In a series of studies, *V. amurensis* has been found to be resistant to *C. gloeosporioides* due to the presence of a resistance-related quantitative trait locus, and due to the expression of a biosynthesis regulator of proanthocyanidins and anthocyanin. Furthermore, it is becoming a potential resistant genetic resource for improving *V. vinifera* grapes [93,94,95]. In East Asia, several table grapes and grapevine breeding materials have been identified as having moderate-to-high levels of resistance to grape ripe rot disease. Instances of interspecific breeding between *V. vinifera* and *V. labruscana*, such as the cultivars ‘Shine Muscat’ and ‘Oriental Star’, are considered moderately tolerant/resistant [92], while the cultivars ‘Agawan’, ‘Hongqi te zao meigui’, ‘Huangguan’, ‘Seosa’, ‘Seyve-Villard 18-315’, ‘Xiangfei 1’, ‘Xiangfei 2’, and ‘Zexiang’ have been found to be resistant to both *C. gloeosporioides* and *C. acutatum* [90]. A recent study using disease resistance molecular markers has identified many grapevine varieties, such as ‘Bailey Alicante A’ and ‘Muscat Bailey A’, as strongly resistant cultivars to both ripe rot pathogens [96].

### 7.2. Rain Sheltering and Bagging as Adequate Cultivation Techniques

Rainwater plays a crucial role in disseminating *Colletotrichum* spp. [5,19,97,98]. Studies have shown that cultivation methods that protect grapes from rainwater through the use of rain shelters and the cluster bagging technique can efficiently decrease the grape ripe rot disease caused by the CGSC [4,99,100]. Du et al. (2015) [99] noted that grapevine rain shelters—by blocking rainfall, reducing leaf wetness, and canopy humidity—significantly decreased ripe rot disease severity and boosted grape yields, enhancing farmers’ income. Consistent with these observations, a grape microflora dynamics analysis indicated that the abundance of *Colletotrichum* was reduced under rain shelter management [101]. Regarding the preharvest bagging of grape clusters, Cosseboom et al. (2022) [4] found that the ripe rot infections caused by the CGSC—including *C. aenigma*, *C. fructicola*, *C. siamense*, and *C. temperatum*, as well as other species in the CASC—were statistically lower in clusters with a bagged treatment for a majority of the season after blooming when compared to those exposed throughout the season. In addition, bagged clusters tended to weigh more than those exposed throughout the season [4]. As found in Liu et al. (2016) [100], the treatment of bagging grape clusters significantly reduced the incidence of ripe rots. Additionally, bagging grape clusters early after the bloom stage demonstrated a decreased *C. gloeosporioides* infection when compared to late-season bagging [100]. Besides the grape ripe rot, the prevention measures used to avoid rain splash can also mitigate other grape diseases. Rain shelter cultivation has been found to decrease grape downy mildew (*Plasmopara viticola*) [102,103] and other grape diseases such as white rot [102], whereas the early preharvest bagging of grape clusters is known to reduce black mold (*Aspergillus niger*), gray mold (*Botrytis cinerea*), powdery mildew (*Uncinula necator*) [104], black rot (*Guignardia bidwellii*), and downy mildew [4]. This implies that rain sheltering and preharvest bagging can be adequate for cultivation measures in managing the complexes of grape cluster rot diseases [3] that are dispersed through rain splash.

### 7.3. Sanitation Practice

Sanitation is an essential measure for reducing primary inoculum and in preventing the spread of plant pathogens [88,105]. In the vineyard, a high percentage of *C. gloeosporioides s.l.* sporulation was found on grape mummies [69]. Ji et al. (2021) [71] used an epidemiological model to show that removing mummies from the previous season resulted in a decrease in inoculum and mitigated grape ripe rot. In an experimental study by Leles et al. (2022) [106], the incidence of ripe rot was significantly lower in the blocks with mummified bunches that were removed in the previous season (18.7%) than in the control blocks where the mummified bunches were not removed (71.2%). It was noted that sanitation practice can be a critical component of integrated management for grape diseases [107,108,109,110].

### 7.4. Sustaining a Healthy Grape Microbiome

Many recent studies have uncovered the diversity of the grapevine microbiome [4], but only a few have reported the presence of *Colletotrichum* spp. [101,111,112]. It is likely that the microbial diversity found in grapes contributes to sustaining plant health. In a study examining the epiphytic fungal communities of grape berries at two different harvest seasons, Ding et al. (2019) [111] found that *Colletotrichum* was one of the dominant genera on grapes during the winter harvest periods but was almost unnoticeable during the summer harvest periods, implying that the abundance of *Colletotrichum* can be significantly influenced by the environments between seasons. During the summer harvest, the principal epiphytic fungal communities on grapes were *Cladosporium*, *Gyrothrix*, *Paramycosphaerella*, *Acremonium*, *Penicillium*, and *Tilletiopsis*. The study by Huang et al. (2022) [101] revealed that grapevines in China were associated with *Colletotrichum*, which was most abundant on grape berries, followed by leaves, rhizosphere soil, and branches. Furthermore, the study showed a reduced abundance of *Colletotrichum* under rain–shelter conditions, and a higher ratio of the fungal genus *Cladosporium* was found in grapevine tissues and rhizosphere soil under rain shelters during various grape growth stages before harvest. In commercial nursery fields in Spain, Gramaje et al. (2019) [112] detected a relatively low abundance of *Colletotrichum* in the woody tissues of grapevine rootstocks, while the main fungal communities were *Cadophora*, *Cladosporium*, *Penicillium*, and *Alternaria*. A genus associated with potential biocontrol activity against fungal pathogens, *Aureobasidium*, was discovered in grapevine wood samples in that study.

Many reports have identified several groups of beneficial microorganisms in grapevine microbiota. *Aureobasidium*, specifically the species *A. pullulans*, has been frequently detected in the grapevine microbiome [112,113,114,115,116,117], and is known to have biocontrol potential against *Colletotrichum* and other pathogens [115,118]. Other fungi in the grape microbiome with possible antagonistic activities and positive effects on plant growth include *Bulleromyces*, *Dioszegia*, *Sporobolomyces*, *Candida* [115], *Alternaria*, *Cladosporium*, *Epicoccum nigrum*, *Trichoderma caerulescens*, *T. gamsii*, and *T. paraviridescens* [113]. In addition, some bacterial antagonists against fungal pathogens have also been found to be associated with the aerial parts of grapevine in microbiome research, such as *Pseudomonas*, *Bacillus*, *Serratia*, *Pantoea*, *Actinomycetes*, *Streptococcus*, and *Burkholderia* [115]. Recent reviews of the grapevine endophytic microbiome have identified several prokaryotic endophytes that contribute to biotic stress tolerance against fungal infections, including *Acinetobacter lwoffii*, *Pseudomonas fluorescens*, *P. migulae*, *Pantoea agglomerans*, *Bacillus subtilis*, *B. pumilus*, *Burkholderia phytofirmans*, *Microbacterium imperiale*, *Kocuria erythromyxa*, *Terribacillus saccharophilus*, *Streptomyces anulatus*, and *Paenibacillus* sp. [115,119]. Given the biodiversity of the grapevine microbiota, it is crucial to maintain a balanced microbial diversity to sustain plant health in grape production systems. Further studies, employing metabarcoding and -omic technologies [120], are needed to uncover the relationships between these beneficial microorganisms and the targeted ripe rot pathogen.

### 7.5. Alternative Biological and Non-Chemical Measures

The management of grape ripe rot disease can feasibly be achieved through the integrated application of biological control agents (BCAs) and other non-chemical materials. Numerous BCAs have been utilized to combat the ripe rot associated with *C. gloeosporioides* (Table 2). The bacterial genus *Bacillus* harbors several potent BCAs useful in controlling grape ripe rot, such as *B. subtilis* [121,122], *B. amyloliquefaciens* [123], *B. licheniformis*, *B. cereus*, *B. aerius*, and *B. velezensis* [122]. The suppression of the grape ripe rot pathogen by *Bacillus* spp. has been demonstrated in vitro [121,122,123,124], as well as in vivo on detached berries [124], detached leaves [122], potted plants [122], and in the field [122,123,124,125]. Nonetheless, in some cases, the application of BCAs did not result in a decrease in ripe rot infection in vineyards [106,121]. In addition, antagonistic fungi and their derivatives can potentially be used to combat grape ripe rot disease. The crude extracts from *Chaetomium cupreum*, *C. globosum*, *Trichoderma harzianum*, *T. hamatum*, and *Penicillium chrysogenum* have been indicated to inhibit the growth of *C. gloeosporioides* isolated from grapes [41], while the application of *Saccharomyces cerevisiae* has shown controlled effects against *C. gloeosporioides* on artificially inoculated grape tissues [126]. Furthermore, *B. subtilis*, *B. amyloliquefaciens*, and *T. harzianum* also exhibited inhibition against the isolates of *C. gloeosporioides* that cause anthracnose-type symptoms in grapes [127,128].

In terms of other nontraditional or non-chemical plant protection materials, chitosan and essential oils may have the potential to inhibit the ripe rot pathogen in grapes [129,130], but potassium phosphite—which is prepared as a neutralized phosphorous acid solution and is helpful in controlling the downy mildew and powdery mildew of grapes—may have no effect on grape ripe rot incidence according to the results of studies that were conducted in the field [106,131]. Lime sulfur was used in order to reduce the overwintered inoculum of the ripe rot pathogen [71], but the application of lime sulfur did not efficiently decrease the incidence of the grape ripe rot disease [106].

### 7.6. Fungicide Selection

Fungicide application to control grape diseases is the norm in most grape-growing areas [132]. Although various integrated methods have been developed to reduce infection, chemical control remains a significant way through which to manage the diseases caused by *Colletotrichum* spp. [19]. There are eight single-site mode-of-action groups used for controlling grape ripe rot disease according to the Fungicide Resistance Action Committee (FRAC) (https://www.frac.info/, accesed on 8 July 2023): B1, methyl benzimidazole carbamates (MBC fungicides) (FRAC 1); B2, N-phenyl carbamates (NPC fungicides) (FRAC 10); C2, succinate dehydrogenase inhibitors (SDHI fungicides) (FRAC 7); C3, quinone outside inhibitors (QoI fungicides) (FRAC 11); D1, anilino-pyrimidine (AP fungicides) (FRAC 9); E2, phenylpyrroles (PP fungicides) (FRAC 12); G1, demethylation inhibitors (DMI fungicides) (FRAC 3); and H4, peptidyl pyrimidine nucleoside (FRAC 19). Table 3 summarizes the recent studies on the use of these fungicides in controlling grape ripe rot disease. 

In targeting cytoskeleton and motor proteins (target site B), MBC fungicides—including benomyl, carbendazim, and thiophanate-methyl—have been widely used [69,127,133,134]. In the wine grape industry of Australia, *C. gloeosporioides* exhibited higher sensitivity to benomyl than *C. acutatum* [6]. Additionally, in Korea, the NPC fungicide diethofencarb has been commonly applied as a mixture with carbendazim [134]. 

In targeting respiration (target site C), SDHI fungicides have been used to control grape ripe rot disease, including benzovindiflupyr, fluxapyroxad, penthiopyrad, fluopyram, boscalid, and pydiflumetofen [19,135]. While some of them were found to be less effective against *C. gloeosporioides* and were applied as a mixture with other fungicides, benzovindiflupyr and penthiopyrad showed high inhibitory activity against *C. gloeosporioides* isolates from grapevine and other anthracnose-causing pathogens that were caused by *Colletotrichum* spp. [135]. QoI fungicides, including azoxystrobin, pyraclostrobin, and trifloxystrobin, have also been used to prevent grape ripe rots, with azoxystrobin and pyraclostrobin being applied more extensively in different regions [19,100,136]. A mixture of pyraclostrobin and boscalid were registered as a commercial product for managing grape ripe rot [19].

In targeting amino acid and protein synthesis (target site D), as well as signal transduction (target site E), the AP fungicide cyprodinil represents a component of the grower standard spray program for controlling grape ripe rot disease in the U.S. [137]. Cyprodinil could be applied in combination with other fungicides [137,138], while the PP fungicide fludioxonil has also been registered as a mixture in the U.S. for grape ripe rot management [19].

In targeting sterol biosynthesis in membranes (target site G) and cell wall biosynthesis (target site H), several DMI fungicides—including prochloraz, difenoconazole, tebuconazole, and triadimenol—have been used to control grape diseases [6,100,133,137,138,139]. Prochloraz, difenoconazole, and tebuconazole are commonly employed fungicides against grape ripe rot in certain grape-growing regions [80,100,133,139]. Greer et al. (2011) [6] investigated the fungicide sensitivity among *Colletotrichum* species and found that the grape ripe rot isolates of *C. acutatum* exhibited significantly greater sensitivity to triadimenol when compared to the isolates of *C. gloeosporioides*. Polyoxins, the peptidyl pyrimidine nucleosides, have been registered for controlling grape ripe rot disease in the U.S. and Taiwan [19,80]. Polyoxins were also reported to effectively inhibit the conidial germination of *C. viniferum* in vitro [80].

In addition to the fungicides categorized under single-site chemistries, some fungicides used for grape disease management have multisite contact activity. Fungicides with multisite activity used for the management of grape ripe rot include oxine-copper, mancozeb, maneb, metiram, thiram, ziram, captan, captafol, folpet, chlorothalonil, iminoctadine, and dithianon [19,69,80,137,140]. In the study of Greer et al. (2011) [6], *C. acutatum* exhibited a higher sensitivity to captan than *C. gloeosporioides* in Australia. Recent in vitro studies have demonstrated that substances such as oxine-copper, mancozeb, metiram, thiram, chlorothalonil, iminoctadine, and dithianon effectively inhibit the grape isolates of the CGSC [80,140].

**Table 3 plants-12-02873-t003:** Fungicides commonly used to control the *Colletotrichum gloeosporioides s.l.* species complex members involved in grape ripe rot disease.

Target Site/FRAC Code	Active Ingredient	Fungicide Resistance Reported	Use against the Ripe Rot Pathogen	Reference
(B1) tubulin polymerization/1	Benomyl	-	Used in the U.S. Previously used in Australia. Reduction in the mycelial growth and sporulation of the isolates from grape. *Colletotrichum gloeosporioides* exhibited a higher sensitivity to benomyl than *C. acutatum*.	[6,69,140]
	Carbendazim	In the field	Widely used to control grape ripe rot in Korea, India, and China. Reduction in the mycelial growth and sporulation of the isolates from grape. Resistant isolates of *C. gloeosporioides* (12/18) were found at the site where the fungicides had been used in Korea. Resistant *C. gloeosporioides* isolates from grapes were prevalent in a study in India.	[127,133,134,140,141]
	Thiophanate-methyl	In the field	Used to control grape ripe rot in China. Has a 37% resistance frequency in a study in China.	[133]
(B2) tubulin polymerization/10	Diethofencarb	In the field	Mixed with carbendazim to control ripe rot in Korea. Resistant isolates of *C. gloeosporioides* (12/18) were found at the site where the fungicides had been used.	[133,134]
(C2) complex II: succinate-dehydrogenase/7	Benzovindiflupyr	-	Highly effective against *C. gloeosporioides* isolates from grapevine in vitro.	[135]
	Fluxapyroxad	-	*C. gloeosporioides* isolates from grapevine in Japan were insensitive to fluxapyroxad.	[135]
	Penthiopyrad	-	Registered for controlling grape ripe rot in Japan. Highly effective against the *C. gloeosporioides* isolates from grapevine.	[135]
	Fluopyram	-	*C. gloeosporioides* isolates from grapevine in Japan were insensitive to fluopyram. Used in combination with tebuconazole in the grower standard spray program for control of grape ripe rot in the U.S.	[135,137,138]
	Boscalid	-	Mild suppression of the grape ripe rot pathogen in vitro. Registered as a mixture with pyraclostrobin for the management of diseases caused by *Colletotrichum* spp. on grapes in the U.S. *C. gloeosporioides* isolates from grapevine in Japan were insensitive to boscalid.	[6,19,135]
	Pydiflumetofen	-	Registered as a mixture with fludioxonil for management of diseases caused by *Colletotrichum* spp. on grape in the U.S.	[19]
(C3) complex III: cytochrome bc1 at Qo site/11	Azoxystrobin	In the field	Inhibition of the grape ripe rot pathogen in vitro. Extensively applied to control grape ripe rot in China. Shows a 97% resistant frequency in *C. gloeosporioides* in a study in China. Isolated with cross-resistance to pyraclostrobin and other quinone outside inhibitors found in the field.	[6,136]
	Pyraclostrobin	Isolates from the field cross-resistance to other quinone outside inhibitors	Inhibition of the grape ripe rot pathogen in vitro. Frequently used in preventing grape ripe rot in Taiwan. Registered as a mixture with boscalid for the management of diseases caused by *Colletotrichum* spp. on grapes in the U.S. Isolated with cross-resistance to azoxystrobin and other quinone outside inhibitors found in the field.	[6,19,100,136,142]
	Trifloxystrobin	-	Inhibition of the grape ripe rot pathogen in vitro.	[6]
(D1) methionine biosynthesis (proposed)/9	Cyprodinil	-	Used in the grower standard spray program for the control of grape ripe rot in the U.S.	[137,138]
(E2) MAP/Histidine-Kinase in osmotic signal transduction/12	Fludioxonil	-	Registered as a mixture with pydiflumetofen for the management of diseases caused by *Colletotrichum* spp. on grapes in the U.S.	[19]
(G1) C14-demethylase in sterol biosynthesis/3	Prochloraz	Slightly resistance generated in the lab	Frequently used in preventing grape ripe rot in Taiwan. Registered for use in controlling grape ripe rot in China. Effectively inhibited the conidial germination and the mycelial growth of *C. viniferum* in vitro. The baseline sensitivity of *C. gloeosporioides* isolates were determined. The isolates from grapes showed a lower sensitivity to prochloraz than the isolates from strawberry.	[80,100,133,142]
	Difenoconazole	In the field	Intensively used to control grape ripe rot in China. Showed a 65.2% resistant frequency in *C. gloeosporioides* in a study in China.	[139]
	Tebuconazole	Slight resistance generated in the laboratory; in the field	Frequently used in preventing grape ripe rot in Taiwan. Registered for use to control grape ripe rot in China. Used in combination with fluopyram in the grower standard spray program to control grapes’ ripe rot in the U.S. The baseline sensitivity of *C. gloeosporioides* isolates was determined. The isolates from grapes showed a lower sensitivity to tebuconazole than the isolates from strawberries. Approximately 30% of the *C. gloeosporioides* showed low-level resistances to tebuconazole in a study in China.	[100,133,137,138,142]
	Triadimenol	-	*C. acutatum* exhibited a higher sensitivity to triadimenol than *C. gloeosporioides* in Australia.	[6]
(H4) chitin synthase/19	Polyoxin	-	Registered for the management of diseases caused by *Colletotrichum* spp. on grape in the U.S. Used to control grape ripe rot in Taiwan. Effectively inhibited the conidial germination of *C. viniferum* in vitro.	[19,80]
(U) cell membrane disruption (proposed)/U12	Dodine	-	Reduction in the mycelial growth and sporulation of the isolates from grape in vitro.	[140]
(M) multisite contact activity/M01	Oxine-copper	-	Used to control grape ripe rot in Taiwan. Effectively inhibited the conidial germination of *C. viniferum* in vitro.	[80]
(M) multisite contact activity/M03	Mancozeb	-	Used to control grape ripe rot in Taiwan. Effectively inhibited the conidial germination of *C. viniferum* in vitro.	[80]
	Maneb	-	Suppressed the level of ripe rot in the field when applied every two weeks from bloom until near harvest in the U.S.	[69]
	Metiram	-	Used to control grape ripe rot in Taiwan. Effectively inhibited the conidial germination of *C. viniferum* in vitro.	[80]
	Thiram	-	Used to control grape ripe rot in Taiwan. Effectively inhibited the conidial germination of *C. viniferum* in vitro.	[80]
	Ziram	-	Registered for the management of diseases caused by *Colletotrichum* spp. on grapes in the U.S.	[19]
(M) multisite contact activity/M04	Captan	-	Significant reduction in ripe rot in the field when applied every two weeks from bloom until near harvest in the U.S. Frequently applied in the grower standard spray program in the control of ripe rot of grapes in the U.S. *C. acutatum* exhibited a higher sensitivity to captan than *C. gloeosporioides* in Australia.	[6,69,137]
	Captafol	-	Significant reduction in ripe rot in the field when applied every two weeks from bloom until near harvest in the U.S.	[69]
	Folpet	-	Significant reduction in ripe rot in the field when applied every two weeks from bloom until near harvest in the U.S.	[69]
(M) multisite contact activity/M05	Chlorothalonil	-	Significant reduction in the mycelial growth and sporulation of the *C. gloeosporioides* isolate from grape in vitro.	[140]
(M) multisite contact activity/M07	Iminoctadine	-	Used to control grape ripe rot in Taiwan. Effectively inhibited the conidial germination and mycelial growth of *C. viniferum* in vitro.	[80]
(M) multisite contact activity/M09	Dithianon	-	Used to control grape ripe rot in Taiwan. Effectively inhibited the conidial germination of *C. viniferum* in vitro.	[80]

### 7.7. Resistance to Fungicides

Addressing fungicide resistance is important for the management of diseases caused by *Colletotrichum* [19]. The reports of the fungicide resistance in the pathogen populations of grape ripe rot disease are summarized in Table 3. Resistance to MBC, NPC, QoI, and DMI fungicides was developed in the *C. gloeosporioides* from grapes.

Resistance to MBC fungicides, including carbendazim and thiophanate-methyl, as well as the NPC fungicide diethofencarb has been observed in Asia. Hwang et al. (2010) [134] established a correlation between the emergence of the carbendazim and diethofencarb-resistant isolates of *C. gloeosporioides* and the regional fungicide application history in Korea. A dual resistance to carbendazim and diethofencarb was also discovered in the grape ripe rot *C. gloeosporioides* isolates that are resistant to thiophanate-methyl in China [133]. Resistance to the MBC and NPC fungicides could be conferred by substitution from tyrosine to phenylalanine in codon 200 of the nuclear β-tubulin gene [133,134]. Interestingly, the behavior of *C. gloeosporioides* isolates that cause anthracnose in grapes displayed further complexity: despite a regional prevalence of carbendazim resistance, these grape isolates remain sensitive to other fungicides from other chemical groups, such as QoI and DMI fungicides [141].

Resistance to QoI fungicides in the grape isolates of *C. gloeosporioides* was documented in China. A few kresoxim-methyl-resistant mutants from the grape isolates were generated after UV light treatments in vitro. The QoI-resistant mutants also showed cross-resistance to pyraclostrobin but not to boscalid, which is an SDHI fungicide, and other DMI fungicides [142]. Another study noted that 62 of the 64 isolates of *C. gloeosporioides* from commercial vineyards were resistant to azoxystrobin. Cross-resistance was observed between azoxystrobin and other QoI fungicides, such as kresoxim-methyl and pyraclostrobin. Resistance was conferred by a substitution from phenylalanine to leucine at the 129th codon, or from glycine to alanine at the 143rd codon in the cytochrome b gene [136].

DMI fungicides have been registered to control grape ripe rot disease worldwide [100,133,137,138,142]. Despite their extensive usage, DMI fungicides have shown a potentially lower risk of resistance development in *C. gloeosporioides* than QoI fungicides. This was revealed by the significantly reduced occurrence of prochloraz and tebuconazole-resistant mutants in contrast to the mutants that are resistant to kresoxim-methyl, which were induced by using UV mutagenesis in vitro [142]. Regarding the mutants from the grape isolate, positive cross-resistance was observed between tebuconazole and difenoconazole, but no cross-resistance was found between tebuconazole and prochloraz or between prochloraz and difenoconazole [142]. In a study by Wang et al. (2020) [139], difenoconazole-resistant isolates from commercial vineyards were discovered with a resistance frequency of 65.2%. Positive cross-resistance was detected between difenoconazole and propiconazole, but not between difenoconazole and prochloraz. Mutations in the sterol 14α-demethylases (CYP51A and CYP51B) gene conferred DMI fungicide resistance to the *C. gloeosporioides* from grapes [139]. Mutations associated with fungicide resistance in *Colletotrichum* were documented in Cortaga et al. (2023) [143].

## 8. Integrated Management of Grape Ripe Rot Disease

Based on our current knowledge of managing the grape ripe rot disease caused by the CGSC, it is crucial to implement integrated strategies that promptly combine preventive and suppressive measures. One primary approach involves selecting, breeding, and utilizing grape cultivars that are resistant to ripe rot to mitigate the diseases. Good preventive cultural measures such as rain sheltering, cluster bagging techniques, and sanitation practices have been shown to significantly reduce inoculum and lower the infection rate of grape ripe rot [4,71,99,100,101,104,106]. Incorporating non-chemical and biological control agents, as well as other integrated measures may have an effect on suppressing grape ripe rot pathogens and sustaining grape health [101,112,113,115,122,123,124,125,129,130]. 

Additionally, the reasonable use of fungicides plays an important role in combating grape ripe rot. Knowing the sensitivity differences of fungicides can assist in planning effective management schemes [19]. Strategies such as the alternate application and combined application of fungicides with different modes of action could help reduce the risk of fungicide resistance among grape ripe rot pathogens [143]. The appropriate combination of fungicides may also enhance efficacy through synergistic effects for disease management [144]. 

Taking an integrated approach, combining fungicide application with cluster bagging has been proven to control grape ripe rot disease effectively and may reduce chemical inputs [4,100,104]. The timely application of fungicides during grape production—whether before bagging [100], in the late season [4,137], or based on the risk of infection [4,11,71]—can be an effective method against the grape ripe rot caused by the CGSC, as well as by CASC and others.

## 9. Conclusions

Grape ripe rot that is caused by the *Colletotrichum* species has been recognized as a major destructive disease in viticulture, leading to significant losses and impacts on both the fresh fruit market and the wine industry. To date, at least 15 species have been discovered within this genus [24], with the CGSC and CASC being the primary groups responsible for grape ripe rot (Table 1). Given the severity of the disease caused by the CGSC in subtropical regions, and its extensive presence globally, this review has primarily focused on the CGSC. However, aspects relating to the CASC have also been discussed to provide a comprehensive view of the grape ripe rot pathology.

This review has outlined the most crucial phytopathological characteristics, as well as the symptom types, life cycle and infection process, hosts, and the population dynamics of the CGSC. Furthermore, it proposes control measures to be incorporated into integrated disease management. However, the disease cycle of grape ripe rot still needs to be further elucidated. This is particularly relevant since the primary and secondary inoculum sources of individual *Colletotrichum* spp. might vary, and the role of overwintering structures in the field is poorly understood. Moreover, studies in other pathosystems have demonstrated that factors such as insects [145], light intensity, the skin thickness of host varieties [146], and nitrogen utilization [147] can significantly influence disease progression. The impact of these factors on the grape–*Colletotrichum* interaction remains unclear, emphasizing the need for further research. We anticipate that after fully understanding the infestation behavior, ecological characteristics, and fungicide sensitivity of the CGSC, it will enable us to devise various strategies to reduce grape ripe rot disease, thus ultimately benefiting the grape industry.

In our opinion, further studies are necessary to clarify the geographical distribution and diversity of the causal agents of grape ripe rot, especially those concerning emerging higher virulence pathogens such as *C. viniferum*. In addition, understanding the CGSC genotypes resistant to fungicides will aid in selecting and applying suitable fungicides or implementing alternative measures in disease control strategies. Although disease prevention and suppression strategies have been discussed herein, we emphasize that implementing thorough sanitation practices and the preharvest bagging of grape clusters are the most critical components of an integrated management strategy for grape ripe rot.

## Figures and Tables

**Figure 1 plants-12-02873-f001:**
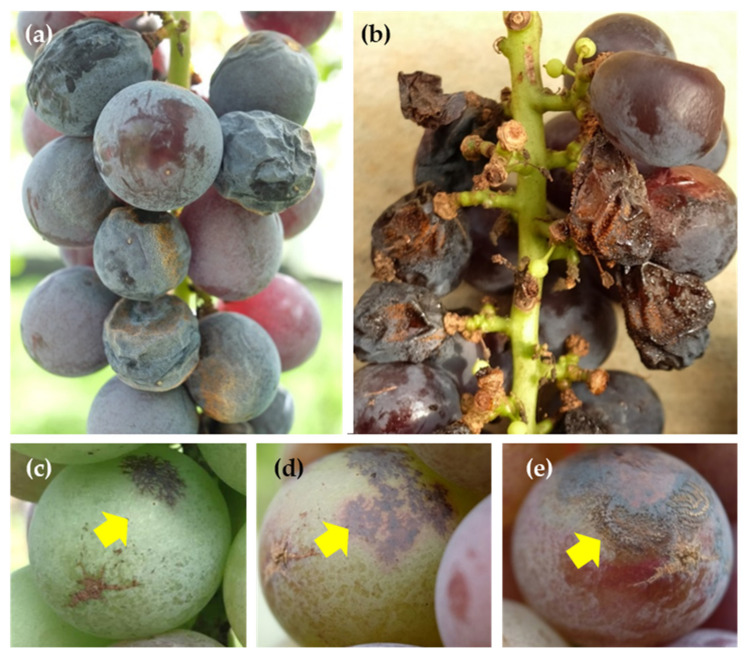
Grape ripe rot on grape berries (cv. Kyoho) caused by *Colletotrichum viniferum* on berries: (**a**,**b**) Ripe rot naturally occurring in the field. (**c**) A single berry inoculated with *C. viniferum* that was captured at the pre-veraison stage, displaying dotted, blocky, or net-like purple brown lesions. (**d**) At veraison, the lesions gradually expanded, developing a purplish red halo. (**e**) At post-veraison, the symptoms evolved into dark, sunken necrotic spots with concentric rings that produce acervuli. Arrows indicate the same infected grape berry at different points in time. Adapted from Lin et al. 2022 [8].

**Figure 2 plants-12-02873-f002:**
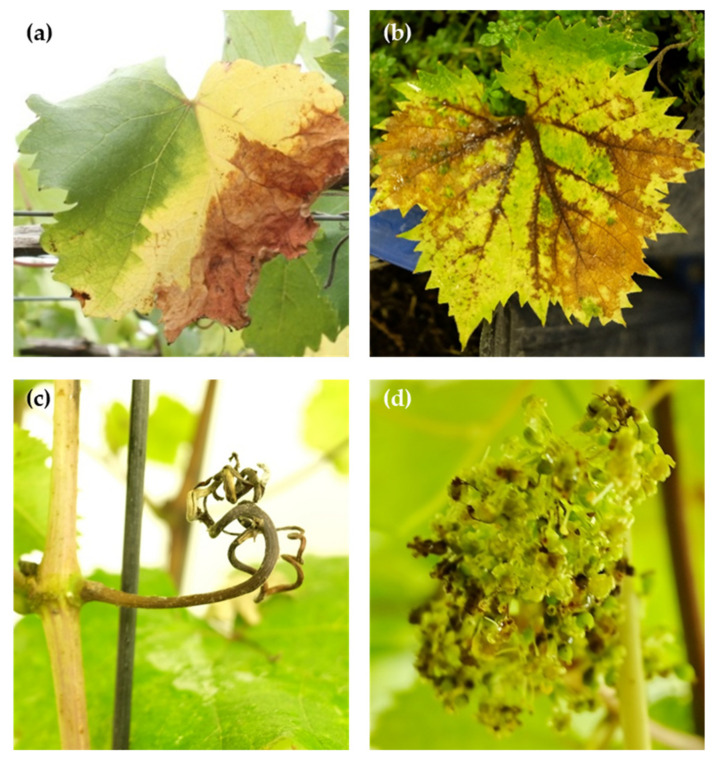
Grape ripe rot on the grape leaves (**a**,**b**), tendrils (**c**), and flowers (**d**) of grapevine (cv. Kyoho) that have been infected by *C. viniferum*; naturally occurring in a, while artificially inoculated in (**b**–**d**). Adapted from Lin et al. 2022 [8].

**Table 1 plants-12-02873-t001:** List of the *Colletotrichum* species documented as causing grape ripe rot.

*Colletotrichum* Species Complex	Species	Country/ Region	Reference
*C. gloeosporioides* complex	*C. aenigma*	China	[30,31]
		Japan	[32] ^2^
		South Korea	[33] ^3^
		USA	[4,12] ^2^
	*C. fructicola*	Brazil	[5]
		China	[9,31]
		Japan	[34] ^2^
		Taiwan	[35]
		USA	[4] ^2^
	*C. gloeosporioides s.s.*	China	[9]
		USA	[12] ^2^
		Japan	[34]
	*C. hebeiense*	China	[30]
	*C. kahawae*	Brazil	[5]
		USA	[12] ^2^
	*C. perseae*	Japan	[34]
	*C. siamense*	Brazil	[36]
		USA	[4] ^2^
	*C. tropicale*	Taiwan	[35,37]
	*C. viniferum*	Brazil	[5]
		China	[9,30,38]
		Japan	[32,34] ^2^
		South Korea	[39]
		Taiwan	[8,40]
	*C. viniferum*-like species (Clade V) ^1^	Japan	[34]
	Other *C. gloeosporioides s.l.*	Australia	[6,16]
		Thailand	[41] ^3^
*C. acutatum* complex	*C. citri*	China	[38]
	*C. fioriniae* ^1^	USA	[4,11,12]
	*C. godetiae*	Italy	[42]
	*C. limitticola*	Brazil	[5]
	*C. nymphaeae*	Brazil	[5]
		USA	[12] ^2^
	*C. pseudoacutatum*	China	[31]
	Other *C. acutatum s.l.*	Australia	[6,16,31,43]
		Japan	[10,44]
		South Korea	[7] ^2^
*C. boninense* complex	*C. karstii*	Brazil	[5]
*C. orchidearum* complex	*C. cliviicola* (syn. *C. cliviae*)-like species	China	[38]

^1^ Although the pathogenicity of these *Colletotrichum* species on grape berries still awaits confirmation through Koch’s postulates, these pathogens are considered the dominant species that cause ripe rot in their respective regions according to related references. Additionally, virulence tests via artificial inoculation have indicated significant pathogenicity. Due to these considerations, they are included in this table. ^2^ While the referenced article reported an isolation from grape berries, Koch’s postulates were not fulfilled in terms of definitively establishing its pathogenicity on berries. However, it is included in this table due to its established role in causing grape ripe rot or because the virulence tests via artificial inoculation that were reported in the reference indicate significant pathogenicity. ^3^ While labeled as anthracnose in the original report, the symptoms described are consistent with ripe rot, and are characterized by berry-rot-type manifestations.

**Table 2 plants-12-02873-t002:** Efficacy of the biological control agents (BCAs) against the *Colletotrichum gloeosporioides s.l.* involved in the grape ripe rot disease.

BCA	Strains/Source	Efficacy against the Ripe Rot Pathogen	Year and Reference
*Bacillus* sp.	M5/ Grape tissue in Taiwan	Inhibited the growth of *C. gloeosporioides* in vitro, lowered the ripe rot disease index on detached fruits, and significantly increased the ratio of intact fruits in a Kyoho grape vineyard.	1993 [124]
*B. subtilis*	KS1/ Grape berry skin of cv. Koshu in Japan.	Suppressed the mycelial growth of *C. gloeosporioides* in vitro, but did not reduce the bunch rot associated with *C. gloeosporioides* in a small-scale test in a Koshu vineyard in the 2008 growing season.	2011 [121]
*B. amyloliquefaciens*	S13-3/ Soil sample from a plum grove in Japan	In vitro inhibition of the mycelial growth of various phytopathogenic fungi, including *C. gloeosporioides*. Significant reduction in the incidence of ripe rot that is caused by *C. gloeosporioides* on the grape berries of *V. vinifera* cv. Semillon in an experimental vineyard.	2012 [123]
*B. cereus*	NRKT/ Unidentified liana in an experimental vineyard in Japan	Significantly reduced the incidence of ripe rot caused by *C. gloeosporioides* on the grape berries of *V. vinifera* cv. Pinot noir in an experimental vineyard.	2017 [72]
*Bacillus* spp.	*B. amyloliquefaciens* D747; *B. subtilis* BV02/ Commercial products	The incidence of grape ripe rot on *Vitis labrusca* was treated with *Bacillus* spp., and this was not different from the control in a field experiment conducted in the 2020 season in Brazil.	2022 [106]
*B. aerius*, *B. velezensis*, and *B. subtilis*	*B. aerius* SB5; *B. velezensis* SB13; *B. subtilis* SC15/ Surface of grape berries in India	Significant inhibition of the mycelial growth of *C. gloeosporioides*. Significant decrease in the infection caused by *C. gloeosporioides* in a detached leaf assay, a potted plants experiment, and in the field, achieved by using the grape cv. Thompson Seedless.	2023 [122]
*Chaetomium*, *Penicillium*, and *Trichoderma*	*Chaetomium cupreum* CC, *C. globosum* CG, *Trichoderma harzianum* PC01, *T. hamatum* PC02, and *Penicillium chrysogenum* KMITL44/ Thailand	The crude extracts from the microbes inhibited the growth of *C. gloeosporioides* from grape. Applications of the bioproducts in a powder formulation in the field reduced ripe rot disease.	2005 [41]
*T. harzianum*	NAIMCC-01965/ Plant pathology laboratory of ICAR-National Research Centre for Grapes in India	Could parasitize *C. gloeosporioides* in vitro and decrease the disease index of the symptoms caused by *C. gloeosporioides* in the artificially inoculated grape leaves of cv. Thompson Seedless.	2012 [127]
*Saccharomyces cerevisiae*	GA8/ Isolated from wine	Significant inhibition of the mycelial growth and conidia germination of *C. gloeosporioides*. Significant control effect against *C. gloeosporioides* in the artificially inoculated grape berries.	2018 [126]

## Data Availability

Not applicable.

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
