# Peer review of "Insights into Grape Ripe Rot: A Focus on the *Colletotrichum gloeosporioides* Species Complex and Its Management Strategies"

_plants, 2023, doi:10.3390/plants12152873_

Round 1

Reviewer 1 Report

This is a well done review of grape ripe rot, and I think is well worth publishing. I did have just one major comment concerning the scope of the paper, as well as other grammatical suggestions and editing comments. 

My biggest concern is about the decision to focus primarily on CGSC and not on CASC. Much of the manuscript covers both species complexes with equivalent importance, only to diverge and focus primarily on CGSC in the latter half of the manuscript. I feel that this narrows the scope of this review too much, and it would be more beneficial to discuss both species complexes. I would consider the scope of the publication to be fine, if other reviews have been written on ripe rot disease of grape. The authors make the claim that the CGSC may be more important than CASC with regards to this disease, yet this is more of a regional variation. They also postulate that CGSC may be more important than CASC in the future, due to climate change. This is a bit speculative, and doesn't consider that regions which previously had cooler temperatures, and little to no ripe rot pressure may begin experiencing ripe rot caused by CASC in the future due to climate change. I also feel that it would not add much length or complexity to the publication to discuss all causative agents of grape ripe rot in the latter half of the paper.

Other primary comments are below. Please find more comments and suggestions in the attached document

P1L28: Has a review of ripe rot been published before? If not, that makes this publication even more important. If one has been previously published, please note how this differs from a prior review of this subject.

P4L107: This is a good figure, but it would be more intuitive to place the A and B images above C, D, and E

P7L304: can you elaborate on cultivation techniques? How can they help with ripe rot management? Cultivation is a word that primarily describes weed control. Perhaps canopy management of the grapevine would be more impactful for grape cluster disease incidence?

P8L307: Has any research been conducted on V. vinifera cultivars? these are the most valuable cultivars for wine production in the world. If not, it may be worth noting that not much research has been conducted on these important varieties

P12L474: It might be best to just discuss fungicides and biologicals that have been tested in the field. Many products show efficacy in vitro or on detached fruit, but do not have efficacy or the same level of efficacy in the field. 

P13L484: There is a LOT of information in this table. I like the column "Fungicide resistance reported". It succinctly gives important information. After reading through this table, I was left wondering if any of these have been tested in efficacy trials in the field. To show this, It would be really nice to have a column similar to "Fungicide resistance reported" called something like "Fungicide Efficacy", with "yes" if a treatment of that product was significantly better than a non-treated control, and "no" if not, and "not tested". I understand that this type of information may not be available, for most of these products. If that is the case, it would be important to make a statement that many of these products have not been directly tested in a proper replicated trial for efficacy against ripe rot. This table takes up 3.5 pages. It has really useful information, but is wordy

P13L485: Same with this table, the in vitro results might be a little too detailed and not accurately represent what will happen in a field scenario

The quality of the English language in this manuscript is excellent. I only caught minor grammatical errors.

Author Response

Dear Reviewer, We sincerely appreciate the time and effort the reviewers have put into evaluating our manuscript. The comments and suggestions provided have greatly helped in enhancing the quality and clarity of our work. In this document, we have systematically responded to each comment raised by you in the table of the attached file. We believe that the revisions made in response to the reviewers' feedback have significantly improved our manuscript. We trust that our responses demonstrate our commitment to creating a valuable contribution to the field and to the readers of your journal. Once again, we express our gratitude for the constructive feedback provided by the reviewers and look forward to hearing from you. Best regards, Chu-Ping Lin

Reviewer 2 Report

The manuscript entitled „Insights into Grape Ripe Rot: A Focus on the Colletotrichum gloeosporioides Species Complex and Its Management Strategies“ provides much information concerning the biology of the pathogen, the significance of this disease to the economy, how to conduct disease management strategies, etc. This paper can have a high impact on the viticulture science community. In my opinion, the manuscript can be accepted in its present form.

Author Response

Dear Reviewer,

We are deeply grateful for your kind words and supportive comments on our manuscript. Your acknowledgment of the potential impact of our work on the viticulture science community is truly encouraging.

Your comments affirm the significance and relevance of our research, which we aimed to make a valuable contribution to the field. Your positive feedback serves as motivation for our continuous effort to enhance our scientific research and its presentation.

Thank you once again for your valuable time and for your supportive comments.

Best regards,

Chu-Ping Lin